# Sorting out typicality with the inverse moment matrix SOS polynomial

**Jean-Bernard Lasserre**
LAAS-CNRS & IMT
Université de Toulouse
31400 Toulouse, France
`lasserre@laas.fr`

**Edouard Pauwels**
IRIT & IMT
Université Toulouse 3 Paul Sabatier
31400 Toulouse, France
`edouard.pauwels@irit.fr`

## Abstract

We study a surprising phenomenon related to the representation of a cloud of data points using polynomials. We start with the previously unnoticed empirical observation that, given a collection (a cloud) of data points, the sublevel sets of a certain distinguished polynomial capture the shape of the cloud very accurately. This distinguished polynomial is a sum-of-squares (SOS) derived in a simple manner from the inverse of the empirical moment matrix. In fact, this SOS polynomial is directly related to orthogonal polynomials and the *Christoffel* function. This allows to generalize and interpret extremality properties of orthogonal polynomials and to provide a mathematical rationale for the observed phenomenon. Among diverse potential applications, we illustrate the relevance of our results on a network intrusion detection task for which we obtain performances similar to existing dedicated methods reported in the literature.

## 1 Introduction

Capturing and summarizing the global shape of a cloud of points is at the heart of many data processing applications such as novelty detection, outlier detection as well as related unsupervised learning tasks such as clustering and density estimation. One of the main difficulties is to account for potentially complicated shapes in multidimensional spaces, or equivalently to account for non standard dependence relations between variables. Such relations become critical in applications, for example in fraud detection where a fraudulent action may be the dishonest combination of several actions, each of them being reasonable when considered on their own.

Accounting for complicated shapes is also related to computational geometry and nonlinear algebra applications, for example integral computation [11] and reconstruction of sets from moments data [6, 7, 12]. Some of these problems have connections and potential applications in machine learning. The work presented in this paper brings together ideas from both disciplines, leading to a method which allows to encode in a simple manner the global shape and spatial concentration of points within a cloud.

We start with a surprising (and apparently unnoticed) empirical observation. Given a collection of points, one may build up a distinguished sum-of-squares (SOS) polynomial whose coefficients (or Gram matrix) is the inverse of the empirical moment matrix (see Section 3). Its degree depends on how many moments are considered, a choice left to the user. Remarkably its sublevel sets capture much of the global shape of the cloud as illustrated in Figure 3. This phenomenon is *not* incidental as illustrated in many additional examples in Appendix A. To the best of our knowledge, this observation has remained unnoticed and the purpose of this paper is to report this empirical finding to the machine learning community and provide first elements toward a mathematical understanding as well as potential machine learning applications.

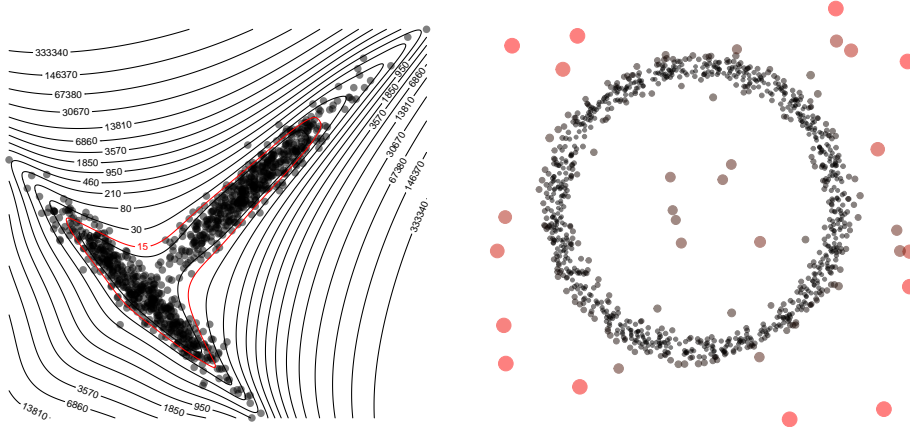

Figure 1: Left: 1000 points in $\mathbb{R}^2$ and the level sets of the corresponding inverse moment matrix SOS polynomial $Q_{\mu,d}$ ($d = 4$). The level set $\binom{p+d}{d}$, which corresponds to the average value of $Q_{\mu,d}$, is represented in red. Right: 1040 points in $\mathbb{R}^2$ with size and color proportional to the value of inverse moment matrix SOS polynomial $Q_{\mu,d}$ ($d = 8$).

The proposed method is based on the computation of the coefficients of a very specific polynomial which depends solely on the empirical moments associated with the data points. From a practical perspective, this can be done via a single pass through the data, or even in an *online* fashion via a sequence of efficient Woodbury updates. Furthermore the computational cost of evaluating the polynomial does *not* depend on the number of data points which is a crucial difference with existing nonparametric methods such as nearest neighbors or kernel based methods [3]. On the other hand, this computation requires the inversion of a matrix whose size depends on the dimension of the problem (see Section 3). Therefore, the proposed framework is suited for moderate dimensions and potentially very large number of observations.

In Section 4 we first describe an affine invariance result which suggests that the distinguished SOS polynomial captures very intrinsic properties of clouds of points. In a second step, we provide a mathematical interpretation that supports our empirical findings based on connections with orthogonal polynomials [5]. We propose a generalization of a well known extremality result for orthogonal univariate polynomials on the real line (or the complex plane) [16, Theorem 3.1.2]. As a consequence, the distinguished SOS polynomial of interest in this paper is understood as the unique optimal solution of a convex optimization problem: minimizing an average value over a structured set of positive polynomials. In addition, we revisit [16, Theorem 3.5.6] about the *Christoffel* function. The mathematics behind provide a simple and intuitive explanation for the phenomenon that we empirically observed.

Finally, in Section 5 we perform numerical experiments on KDD cup network intrusion dataset [13]. Evaluation of the distinguished SOS polynomial provides a score that we use as a measure of outlyingness to detect network intrusions (assuming that they correspond to outlier observations). We refer the reader to [3] for a discussion of available methods for this task. For the sake of a fair comparison we have reproduced the experiments performed in [18] for the same dataset. We report results similar to (and sometimes better than) those described in [18] which suggests that the method is comparable to other dedicated approaches for network intrusion detection, including robust estimation and Mahalanobis distance [8, 10], mixture models [14] and recurrent neural networks [18].

## 2 Multivariate polynomials, moments and sums of squares

**Notations:** We fix the ambient dimension to be $p$ throughout the text. For example, we will manipulate vectors in $\mathbb{R}^p$ as well as $p$-variate polynomials with real coefficients. We denote by $X$ a set of $p$ variables $X_1, \ldots, X_p$ which we will use in mathematical expressions defining polynomials. We identify monomials from the canonical basis of $p$-variate polynomials with their exponents in $\mathbb{N}^p$: we associate to $\alpha = (\alpha_i)_{i=1\ldots p} \in \mathbb{N}^p$ the monomial $X^\alpha := X_1^{\alpha_1} X_2^{\alpha_2} \ldots X_p^{\alpha_p}$ which degree is $\deg(\alpha) := \sum_{i=1}^p \alpha_i$. We use the expressions $<_{gl}$ and $\leq_{gl}$ to denote the graded lexicographic order, a well ordering over $p$-variate monomials. This amounts to, first, use the canonical order on the

degree and, second, break ties in monomials with the same degree using the lexicographic order with $X_1 = a, X_2 = b \ldots$ For example, the monomials in two variables $X_1, X_2$, of degree less or equal to 3 listed in this order are given by: $1, X_1, X_2, X_1^2, X_1 X_2, X_2^2, X_1^3, X_1^2 X_2, X_1 X_2^2, X_2^3$.

We denote by $\mathbb{N}_d^p$, the set $\{\alpha \in \mathbb{N}^p; \deg(\alpha) \leq d\}$ ordered by $\leq_{gl}$. $\mathbb{R}[X]$ denotes the set of $p$-variate polynomials: linear combinations of monomials with real coefficients. The degree of a polynomial is the highest of the degrees of its monomials with nonzero coefficients[1]. We use the same notation, $\deg(\cdot)$, to denote the degree of a polynomial or of an element of $\mathbb{N}^p$. For $d \in \mathbb{N}$, $\mathbb{R}_d[X]$ denotes the set of $p$-variate polynomials of degree less or equal to $d$. We set $s(d) = \binom{p+d}{d}$, the number of monomials of degree less or equal to $d$. We will denote by $\mathbf{v}_d(X)$ the vector of monomials of degree less or equal to $d$ sorted by $\leq_{gl}$. We let $\mathbf{v}_d(X) := (X^\alpha)_{\alpha \in \mathbb{N}_d^p} \in \mathbb{R}_d[X]^{s(d)}$. With this notation, we can write a polynomial $P \in \mathbb{R}_d[X]$ as follows $P(X) = \langle \mathbf{p}, \mathbf{v}_d(X) \rangle$ for some real vector of coefficients $\mathbf{p} = (p_\alpha)_{\alpha \in \mathbb{N}_d^p} \in \mathbb{R}^{s(d)}$ ordered using $\leq_{gl}$. Given $\mathbf{x} = (x_i)_{i=1\ldots p} \in \mathbb{R}^p$, $P(\mathbf{x})$ denotes the evaluation of $P$ with the assignments $X_1 = x_1, X_2 = x_2, \ldots X_p = x_p$. Given a Borel probability measure $\mu$ and $\alpha \in \mathbb{N}^p$, $y_\alpha(\mu)$ denotes the moment $\alpha$ of $\mu$: $y_\alpha(\mu) = \int_{\mathbb{R}^p} \mathbf{x}^\alpha d\mu(\mathbf{x})$. Throughout the paper, we will only consider measures of which all moments are finite.

**Moment matrix:** Given a Borel probability measure $\mu$ on $\mathbb{R}^p$, the moment matrix of $\mu$, $M_d(\mu)$, is a matrix indexed by monomials of degree at most $d$ ordered by $\leq_{gl}$. For $\alpha, \beta \in \mathbb{N}_d^p$, the corresponding entry in $M_d(\mu)$ is defined by $M_d(\mu)_{\alpha,\beta} := y_{\alpha+\beta}(\mu)$, the moment $\alpha + \beta$ of $\mu$. When $p = 2$, letting $y_\alpha = y_\alpha(\mu)$ for $\alpha \in \mathbb{N}_4^2$, we have

$$
M_2(\mu): \quad
\begin{array}{c|cccccc}
 & 1 & X_1 & X_2 & X_1^2 & X_1 X_2 & X_2^2 \\
\hline
1 & 1 & y_{10} & y_{01} & y_{20} & y_{11} & y_{02} \\
X_1 & y_{10} & y_{20} & y_{11} & y_{30} & y_{21} & y_{12} \\
X_2 & y_{01} & y_{11} & y_{02} & y_{21} & y_{12} & y_{03} \\
X_1^2 & y_{20} & y_{30} & y_{21} & y_{40} & y_{31} & y_{22} \\
X_1 X_2 & y_{11} & y_{21} & y_{12} & y_{31} & y_{22} & y_{13} \\
X_2^2 & y_{02} & y_{12} & y_{03} & y_{22} & y_{13} & y_{04}
\end{array} .
$$

$M_d(\mu)$ is positive semidefinite for all $d \in \mathbb{N}$. Indeed, for any $\mathbf{p} \in \mathbb{R}^{s(d)}$, let $P \in \mathbb{R}_d[X]$ be the polynomial with vector of coefficients $\mathbf{p}$, we have $\mathbf{p}^T M_d(\mu)\mathbf{p} = \int_{\mathbb{R}^p} P^2(\mathbf{x})d\mu(\mathbf{x}) \geq 0$. Furthermore, we have the identity $M_d(\mu) = \int_{\mathbb{R}^p} \mathbf{v}_d(\mathbf{x})\mathbf{v}_d(\mathbf{x})^T d\mu(\mathbf{x})$ where the integral is understood elementwise.

**Sum of squares (SOS):** We denote by $\Sigma[X] \subset \mathbb{R}[X]$ (resp. $\Sigma_d[X] \subset \mathbb{R}_d[X]$), the set of polynomials (resp. polynomials of degree at most $d$) which can be written as a sum of squares of polynomials. Let $P \in \mathbb{R}_{2m}[X]$ for some $m \in \mathbb{N}$, then $P$ belongs to $\Sigma_{2m}[X]$ if there exists a finite $J \subset \mathbb{N}$ and a family of polynomials $P_j \in \mathbb{R}_m[X]$, $j \in J$, such that $P = \sum_{j \in J} P_j^2$. It is obvious that sum of squares polynomials are always nonnegative. A further interesting property is that this class of polynomials is connected with positive semidefiniteness. Indeed, $P$ belongs to $\Sigma_{2m}[X]$ if and only if

$$
\exists Q \in \mathbb{R}^{s(m) \times s(m)}, \; Q \succeq 0, \; P(\mathbf{x}) = \mathbf{v}_d(\mathbf{x})^T Q \mathbf{v}_d(\mathbf{x}), \; \forall \mathbf{x} \in \mathbb{R}^p. \tag{1}
$$

As a consequence, every positive semidefinite matrix $Q \in \mathbb{R}^{s(m) \times s(m)}$ defines a polynomial in $\Sigma_{2m}[X]$ by using the representation in (1).

## 3 Empirical observations on the inverse moment matrix SOS polynomial

The *inverse moment-matrix SOS polynomial* is associated to a measure $\mu$ which satisfies the following.

**Assumption 1** *$\mu$ is a Borel probability measure on $\mathbb{R}^p$ with all its moments finite and $M_d(\mu)$ is positive definite for a given $d \in \mathbb{N}$.*

**Definition 1** *Let $\mu, d$ satisfy Assumption 1. We call the SOS polynomial $Q_{\mu,d} \in \Sigma_{2d}[X]$ defined by the application:*

$$
\mathbf{x} \mapsto \quad Q_{\mu,d}(\mathbf{x}) := \mathbf{v}_d(\mathbf{x})^T M_d(\mu)^{-1} \mathbf{v}_d(\mathbf{x}), \qquad \mathbf{x} \in \mathbb{R}^p, \tag{2}
$$

*the inverse moment-matrix SOS polynomial of degree $2d$ associated to $\mu$.*

Actually, connection to orthogonal polynomials will show that the inverse function $\mathbf{x} \mapsto Q_{\mu,d}(\mathbf{x})^{-1}$ is called the *Christoffel* function in the literature [16, 5] (see also Section 4).

In the remainder of this section, we focus on the situation when $\mu$ corresponds to an empirical measure over $n$ points in $\mathbb{R}^p$ which are fixed. So let $\mathbf{x}_1, \dots, \mathbf{x}_n \in \mathbb{R}^p$ be a fixed set of points and let $\mu := \frac{1}{n} \sum_{i=1}^{n} \delta_{\mathbf{x}_i}$ where $\delta_{\mathbf{x}}$ corresponds to the Dirac measure at $\mathbf{x}$. In such a case the polynomial $Q_{\mu,d}$ in (2) is determined only by the empirical moments up to degree $2d$ of our collection of points. Note that we also require that $M_d(\mu) \succ 0$. In other words, the points $\mathbf{x}_1, \dots, \mathbf{x}_n$ do not belong to an algebraic set defined by a polynomial of degree less or equal to $d$. We first describe empirical properties of inverse moment matrix SOS polynomial in this context of empirical measures. A mathematical intuition and further properties behind these observations are developed in Section 4.

## 3.1 Sublevel sets

The starting point of our investigations is the following phenomenon which to the best of our knowledge has remained unnoticed in the literature. For the sake of clarity and simplicity we provide an illustration in the plane. Consider the following experiment in $\mathbb{R}^2$ for a fixed $d \in \mathbb{N}$: represent on the same graphic, the cloud of points $\{\mathbf{x}_i\}_{i=1\dots n}$ and the sublevel sets of SOS polynomial $Q_{\mu,d}$ in $\mathbb{R}^2$ (equivalently, the superlevel sets of the Christoffel function). This is illustrated in the left panel of Figure 3. The collection of points consists of 500 simulations of two different Gaussians and the value of $d$ is 4. The striking feature of this plot is that the level sets capture the global shape of the cloud of points quite accurately. In particular, the level set $\{\mathbf{x} : Q_{\mu,d}(\mathbf{x}) \le \binom{p+d}{d}\}$ captures most of the points. We could reproduce very similar observations on different shapes with various number of points in $\mathbb{R}^2$ and degree $d$ (see Appendix A).

## 3.2 Measuring outlyingness

An additional remark in a similar line is that $Q_{\mu,d}$ tends to take higher values on points which are isolated from other points. Indeed in the left panel of Figure 3, the value of the polynomial tends to be smaller on the boundary of the cloud. This extends to situations where the collection of points correspond to shape with a high density of points with a few additional outliers. We reproduce a similar experiment on the right panel of Figure 3. In this example, 1000 points are sampled close to a ring shape and 40 additional points are sampled uniformly on a larger square. We do not represent the sublevel sets of $Q_{\mu,d}$ here. Instead, the color and shape of the points are taken proportionally to the value of $Q_{\mu,d}$, with $d = 8$.

First, the results confirm the observation of the previous paragraph, points that fall close to the ring shape tend to be smaller and points on the boundary of the ring shape are larger. Second, there is a clear increase in the size of the points that are relatively far away from the ring shape. This highlight the fact that $Q_{\mu,d}$ tends to take higher value in less populated areas of the space.

## 3.3 Relation to maximum likelihood estimation

If we fix $d = 1$, we recover the maximum likelihood estimation for the Gaussian, up to a constant additive factor. To see this, set $\mu = \frac{1}{n} \sum_{i=1}^{n} \mathbf{x}_i$ and $S = \frac{1}{n} \sum_{i=1}^{n} \mathbf{x}_i \mathbf{x}_i^T$. With this notation, we have the following block representation of the moment matrix,

$$M_d(\mu) = \begin{pmatrix} 1 & \mu^T \\ \mu & S \end{pmatrix} \qquad M_d(\mu)^{-1} = \begin{pmatrix} 1 + \mu^T V^{-1} \mu & -\mu^T V^{-1} \\ -V^{-1}\mu & V^{-1} \end{pmatrix},$$

where $V = S - \mu\mu^T$ is the empirical covariance matrix and the expression for the inverse is given by Schur complement. In this case, we have $Q_{\mu,1}(\mathbf{x}) = 1 + (\mathbf{x} - \mu)^T V^{-1} (\mathbf{x} - \mu)$ for all $\mathbf{x} \in \mathbb{R}^p$. We recognize the quadratic form that appears in the density function of the multivariate Gaussian with parameters estimated by maximum likelihood. This suggests a connection between the inverse SOS moment polynomial and maximum likelihood estimation. Unfortunately, this connection is difficult to generalize for higher values of $d$ and we do not pursue the idea of interpreting the empirical observations of this section through the prism of maximum likelihood estimation and leave it for further research. Instead, we propose an alternative view in Section 4.

## 3.4 Computational aspects

Recall that $s(d) = \binom{p+d}{d}$ is the number of $p$-variate monomials of degree up to $d$. The computation of $Q_{\mu,d}$ requires $O(ns(d)^2)$ operations for the computation of the moment matrix and $O(s(d)^3)$ operations for the matrix inversion. The evaluation of $Q_{\mu,d}$ requires $O(s(d)^2)$ operations.

Estimating the coefficients of $Q_{\mu,d}$ has a computational cost that depends only linearly in the number of points $n$. The cost of evaluating $Q_{\mu,d}$ is constant with respect to the number of points $n$. This is an important contrast with kernel based or distance based methods (such as nearest neighbors and one class SVM) for density estimation or outlier detection since they usually require at least $O(n^2)$ operations for the evaluation of the model [3]. Moreover, this is well suited for online settings where inverse moment matrix computation can be done using rank one Woodbury updates [15, Section 2.7.1].

The dependence in the dimension $p$ is of the order of $p^d$ for a fixed $d$. Similarly, the dependence in $d$ is of the order of $d^p$ for a fixed dimension $p$ and the joint dependence is exponential. Furthermore, $M_d(\mu)$ has a Hankel structure which is known to produce ill conditioned matrices. This suggests that the direct computation and evaluation of $Q_{\mu,d}$ will mostly make sense for moderate dimensions and degree $d$. In our experiments, for large $d$, the evaluation of $Q_{\mu,d}$ remains quite stable, but the inversion leads to numerical error for higher values (around 20).

## 4 Invariance and interpretation through orthogonal polynomials

The purpose of this section is to provide a mathematical rationale that *explains* the empirical observations made in Section 3. All the proofs are postponed to Appendix B. We fix a Borel probability measure $\mu$ on $\mathbb{R}^p$ which satisfies Assumption 1. Note that $M_d(\mu)$ is always positive definite if $\mu$ is not supported on the zero set of a polynomial of degree at most $d$. Under Assumption 1, $M_d(\mu)$ induces an inner product on $\mathbb{R}^{s(d)}$ and by extension on $\mathbb{R}_d[X]$ (see Section 2). This inner product is denoted by $\langle \cdot, \cdot \rangle_\mu$ and satisfies for any polynomials $P, Q \in \mathbb{R}_d[X]$ with coefficients $\mathbf{p}, \mathbf{q} \in \mathbb{R}^{s(d)}$,

$$\langle P, Q \rangle_\mu := \langle \mathbf{p}, M_d(\mu)\mathbf{q} \rangle_{\mathbb{R}^{s(d)}} = \int_{\mathbb{R}^p} P(\mathbf{x})Q(\mathbf{x})d\mu(\mathbf{x}).$$

We will also use the canonical inner product over $\mathbb{R}_d[X]$ which we write $\langle P, Q \rangle_{\mathbb{R}_d[X]} := \langle \mathbf{p}, \mathbf{q} \rangle_{\mathbb{R}^{s(d)}}$ for any polynomials $P, Q \in \mathbb{R}_d[X]$ with coefficients $\mathbf{p}, \mathbf{q} \in \mathbb{R}^{s(d)}$. We will omit the subscripts for this canonical inner product and use $\langle \cdot, \cdot \rangle$ for both products.

### 4.1 Affine invariance

It is worth noticing that the mapping $\mathbf{x} \mapsto Q_{\mu,d}(\mathbf{x})$ does not depend on the particular choice of $\mathbf{v}_d(X)$ as a basis of $\mathbb{R}_d[X]$, any other basis would lead to the same mapping. This leads to the result that $Q_{\mu,d}$ captures affine invariant properties of $\mu$.

**Lemma 1** *Let $\mu$ satisfy Assumption 1 and $A \in \mathbb{R}^{p \times p}, b \in \mathbb{R}^p$ define an invertible affine mapping on $\mathbb{R}^p$, $\mathcal{A}\colon \mathbf{x} \to A\mathbf{x}+b$. Then, the push forward measure, defined by $\tilde{\mu}(S) = \mu(\mathcal{A}^{-1}(S))$ for all Borel sets $S \subset \mathbb{R}^p$, satisfies Assumption 1 (with the same $d$ as $\mu$) and for all $\mathbf{x} \in \mathbb{R}^p$, $Q_{\mu,d}(\mathbf{x}) = Q_{\tilde{\mu},d}(A\mathbf{x} + b)$.*

Lemma 1 is probably better understood when $\mu = 1/n \sum_{i=1}^n \delta_{\mathbf{x}_i}$ as in Section 3. In this case, we have $\tilde{\mu} = 1/n \sum_{i=1}^n \delta_{A\mathbf{x}_i+b}$ and Lemma 1 asserts that the level sets of $Q_{\tilde{\mu},d}$ are simply the images of those of $Q_{\mu,d}$ under the affine transformation $\mathbf{x} \mapsto A\mathbf{x} + b$. This is illustrated in Appendix D.

### 4.2 Connection with orthogonal polynomials

We define a classical [16, 5] family of orthonormal polynomials, $\{P_\alpha\}_{\alpha \in \mathbb{N}_d^p}$ ordered according to $\leq_{gl}$ which satisfies for all $\alpha \in \mathbb{N}_d^p$

$$\langle P_\alpha, X^\beta \rangle = 0 \text{ if } \alpha <_{gl} \beta, \ \langle P_\alpha, P_\alpha \rangle_\mu = 1, \ \langle P_\alpha, X^\beta \rangle_\mu = 0 \text{ if } \beta <_{gl} \alpha, \ \langle P_\alpha, X^\alpha \rangle_\mu > 0. \quad (3)$$

It follows from (3) that $\langle P_\alpha, P_\beta \rangle_\mu = 0$ if $\alpha \neq \beta$. Existence and uniqueness of such a family is guaranteed by the Gram-Schmidt orthonormalization process following the $\leq_{gl}$ order, and by the

positivity of the moment matrix, see for instance [5, Theorem 3.1.11]. There exist determinantal formulae [9] and more precise description can be made for measures which have additional geometric properties, see [5] for many examples.

Let $D_d(\mu)$ be the lower triangular matrix whose rows are the coefficients of the polynomials $P_\alpha$ defined in (3) ordered by $\leq_{gl}$. It can be shown that $D_d(\mu) = L_d(\mu)^{-T}$, where $L_d(\mu)$ is the Cholesky factorization of $M_d(\mu)$. Furthermore, there is a direct relation with the inverse moment matrix as $M_d(\mu)^{-1} = D_d(\mu)^T D_d(\mu)$ [9, Proof of Theorem 3.1]. This has the following consequence.

**Lemma 2** *Let $\mu$ satisfy Assumption 1, then $Q_{\mu,d} = \sum_{\alpha \in \mathbb{N}_d^p} P_\alpha^2$, where the family $\{P_\alpha\}_{\alpha \in \mathbb{N}_d^p}$ is defined by (3) and $\int_{\mathbb{R}^p} Q_{\mu,d}(\mathbf{x})d\mu(\mathbf{x}) = s(d)$.*

That is, $Q_{\mu,d}$ is a very specific and distinguished SOS polynomial, the sum of squares of the orthonormal basis elements $\{P_\alpha\}_{\alpha \in \mathbb{N}_d^p}$ of $\mathbb{R}_d(X)$ (w.r.t. $\mu$). Furthermore, the average value of $Q_{\mu,d}$ with respect to $\mu$ is $s(d)$ which corresponds to the red level set in left panel of Figure 3.

### 4.3   A variational formulation for the inverse moment matrix SOS polynomial

In this section, we show that the family of polynomials $\{P_\alpha\}_{\alpha \in \mathbb{N}_d^p}$ defined in (3) is the unique solution (up to a multiplicative constant) of a convex optimization problem over polynomials. This fact combined with Lemma 2 provides a mathematical rationale for the empirical observations outlined in Section 3. Consider the following optimization problem.

$$\min_{Q_\alpha, \theta_\alpha, \alpha \in \mathbb{N}_d^p} \quad \frac{1}{2} \int_{\mathbb{R}^p} \sum_{\alpha \in \mathbb{N}_d^p} Q_\alpha(\mathbf{x})^2 d\mu(\mathbf{x}) \tag{4}$$

$$\text{s.t.} \quad \mathbf{q}_{\alpha\alpha} \geq \exp(\theta_\alpha), \quad \mathbf{q}_{\alpha\beta} = 0, \quad \alpha, \beta \in \mathbb{N}_d^p, \quad \alpha <_{gl} \beta, \quad \sum_{\alpha \in \mathbb{N}_d^p} \theta_\alpha = 0,$$

where $Q_\alpha(\mathbf{x}) = \sum_{\beta \in \mathbb{N}_d^p} \mathbf{q}_{\alpha\beta} \mathbf{x}^\beta$ is a polynomial and $\theta_\alpha$ is a real variable for each $\alpha \in \mathbb{N}_d^p$. We first comment on problem (4). Let $P = \sum_{\alpha \in \mathbb{N}_d^p} Q_\alpha^2$ be the SOS polynomial appearing in the objective function of (4). The objective of (4) simply involves the average value of $P$ with respect to $\mu$. Let $S_d \subset \Sigma_d[X]$ be the set of such SOS polynomials $P$ which have a sum of square decomposition satisfying the constraints of (4) (for some arbitrary value of the real variables $\{\theta_\alpha\}_{\alpha \in \mathbb{N}_d^p}$). With this notation, problem (4) has the simple formulation $\min_{P \in S_d} \frac{1}{2} \int P d\mu$.

Based on this formulation, problem (4) can be interpreted as balancing two antagonist targets. On one hand the minimization of the average value of the SOS polynomial $P$ with respect to $\mu$, on the other hand the avoidance of the trivial polynomial, enforced by the constraint that $P \in S_d$. The constraint $P \in S_d$ is simple and natural. It ensures that $P$ is a sum of squares of polynomials $\{Q_\alpha\}_{\alpha \in \mathbb{N}_d^p}$, where the leading term of each $Q_\alpha$ (according to the ordering $\leq_{gl}$) is $\mathbf{q}_{\alpha\alpha} \mathbf{x}^\alpha$ with $\mathbf{q}_{\alpha\alpha} > 0$ (and hence does not vanish). Inversely, using Cholesky factorization, for any SOS polynomial $Q$ of degree $2d$ which coefficient matrix (see equation (1)) is positive definite, there exists $a > 0$ such that $aQ \in S_d$. This suggests that $S_d$ is a quite general class of nonvanishing SOS polynomials. The following result, which gives a relation between $Q_{\mu,d}$ and solutions of (4), uses a generalization of [16, Theorem 3.1.2] to several orthogonal polynomials of several variables.

**Theorem 1** : *Under Assumption 1, problem (4) is a convex optimization problem with a unique optimal solution $(Q_\alpha^*, \theta_\alpha^*)$, which satisfies $Q_\alpha^* = \sqrt{\lambda} P_\alpha$, $\alpha \in \mathbb{N}_d^p$, for some $\lambda > 0$. In particular, the distinguished SOS polynomial $Q_{\mu,d} = \sum_{\alpha \in \mathbb{N}_d^p} P_\alpha^2 = \frac{1}{\lambda} \sum_{\alpha \in \mathbb{N}_d^p} (Q_\alpha^*)^2$, is (part of) the unique optimal solution of (4).*

Theorem 1 states that up to the scaling factor $\lambda$, the distinguished SOS polynomial $Q_{\mu,d}$ is *the unique optimal solution of problem (4)*. A detailed proof is provided in the Appendix B and we only sketch the main ideas here. First, it is remarkable that for each fixed $\alpha \in \mathbb{N}_d^p$ (and again up to a scaling factor) the polynomial $P_\alpha$ is the unique optimal solution of the problem: $\min_Q \left\{ \int Q^2 d\mu : Q \in \mathbb{R}_d[X], Q(\mathbf{x}) = \mathbf{x}^\alpha + \sum_{\beta <_{gl} \alpha} \mathbf{q}_\beta \mathbf{x}^\beta \right\}$. This fact is well-known in the univariate case [16, Theorem 3.1.2] and does not seem to have been exploited in the literature, at

least for purposes similar to ours. So intuitively, $P_\alpha^2$ should be as close to $0$ as possible on the support of $\mu$. Problem (4) has similar properties and the constraint on the vector of weights $\theta$ enforces that, at an optimal solution, the contribution of $\int (Q_\alpha^*)^2 \, d\mu$ to the overall sum in the criterion is the same for all $\alpha$. Using Lemma 2 yields (up to a multiplicative constant) the polynomial $Q_{\mu,d}$. Other constraints on $\theta$ would yield different weighted sum of the squares $P_\alpha^2$. This will be a subject of further investigations.

To sum up, Theorem 1 provides a rationale for our observations. Indeed when solving (4), intuitively, $Q_{\mu,d}$ should be close to $0$ on average while remaining in a class of nonvanishing SOS polynomials.

### 4.4 Christoffel function and outlier detection

The following result from [5, Theorem 3.5.6] draws a direct connection between $Q_{\mu,d}$ and the *Chritoffel* function (the right hand side of (5)).

**Theorem 2 ([5])** *Let Assumption 1 hold and let* $\mathbf{z} \in \mathbb{R}^p$ *be fixed, arbitrary. Then*

$$Q_{\mu,d}(\mathbf{z})^{-1} = \min_{P \in \mathbb{R}_d[X]} \left\{ \int_{\mathbb{R}^p} P(\mathbf{x})^2 \, d\mu(\mathbf{x}) : P(\mathbf{z}) = 1 \right\}. \tag{5}$$

Theorem 2 provides a mathematical rationale for the use of $Q_{\mu,d}$ for outlier or novelty detection purposes. Indeed, from Lemma 2 and equation (3), we have $Q_{\mu,d} \geq 1$ on $\mathbb{R}^p$. Furthermore, the solution of the minimization problem in (5) satisfies $P(\mathbf{z})^2 = 1$ and $\mu\left(\left\{\mathbf{x} \in \mathbb{R}^p : P(\mathbf{x})^2 \leq 1\right\}\right) \geq 1 - Q_{\mu,d}(\mathbf{z})^{-1}$ (by Markov's inequality). Hence, for high values of $Q_{\mu,d}(\mathbf{z})$, the sublevel set $\left\{\mathbf{x} \in \mathbb{R}^p : P(\mathbf{x})^2 \leq 1\right\}$ contains most of the mass of $\mu$ while $P(\mathbf{z})^2 = 1$. An illustration of this discussion is given in appendix E. Again the result of Theorem 2 does not seem to have been interpreted for purposes similar to ours.

## 5   Experiments on network intrusion datasets

In addition to having its own mathematical interest, Theorem 1 can be exploited for various purposes. For instance, the sub-level sets of $Q_{\mu,d}$, and in particular $\{\mathbf{x} \in \mathbb{R}^p : Q_{\mu,d}(\mathbf{x}) \leq \binom{p+d}{d}\}$, can be used to encode a cloud of points in a simple and compact form. However in this section we focus on another potential application in anomaly detection.

Empirical findings described in Section 3 suggest that the polynomial $Q_{\mu,d}$ can be used to detect outliers in a collection of real vectors (with $\mu$ the empirical average). This is backed up by the results presented in Section 4. We illustrate these properties on a real world example. We choose the KDD cup 99 network intrusion dataset [13] consisting of network connection data, labeled as normal traffic or network intrusions. We follow [19] and [18] and construct five datasets consisting of labeled vectors in $\mathbb{R}^3$ with the following properties

| Dataset | http | smtp | ftp-data | ftp | others |
|---|---|---|---|---|---|
| Number of examples | 567498 | 95156 | 30464 | 4091 | 5858 |
| Proportions of attacks | 0.004 | 0.0003 | 0.023 | 0.077 | 0.016 |

The details on the datasets construction are available in [19, 18] and reproduced in Appendix C. The main idea is to compute an outlyingness score (independant of the label) and compare outliers predicted by the score and network intrusion labels. The underlying assumption is that network intrusions correspond to infrequent abnormal behaviors and could be considered as outliers.

We reproduce the same experiment as in [18, Section 5.4] using the value of $Q_{\mu,d}$ from Definition 1 as an outlyingness score (with $d = 3$). The authors of [18] have compared different methods in the same experimental setting: robust estimation and Mahalanobis distance [8, 10], mixture models [14] and recurrent neural networks. The results are gathered in [18, Figure 7]. In the left panel of Figure 2 we represent the same performance measure for our approach: we first compute the value of $Q_{\mu,d}$ for each datapoint and use it as an outlyingness score. We then display the proportion of correctly identified outliers, with score above a given threshold, as a function of the proportion of examples with score above the threshold (for different values of the threshold). The main comments are as follows.

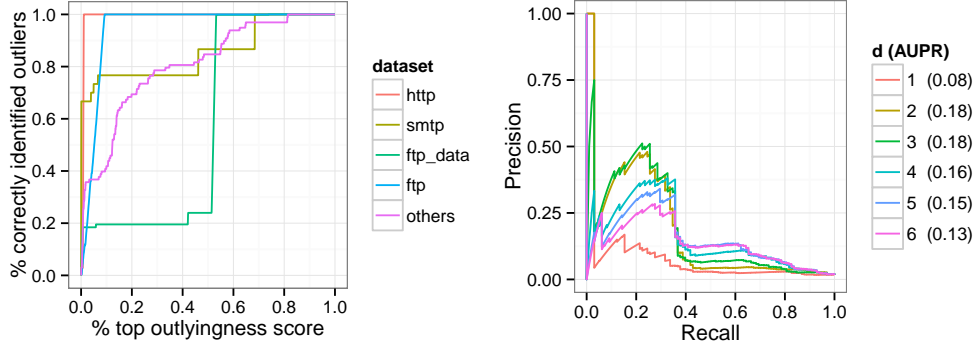

Figure 2: Left: reproduction of the results described in [18] with the evaluation of $Q_{\mu,d}$ as an outlyingness score ($d = 3$). Right: precision-recall curves for different values of $d$ (dataset "others").

• The inverse moment matrix SOS polynomial does detect network intrusions with varying performances on the five datasets.

• Except for the "ftp-data dataset", the global shape of these curves are very similar to results reported in [18, Figure 7] indicating that the proposed approach is comparable to other dedicated methods for intrusion detection in these four datasets.

In a second experiment, we investigate the effect of changing the value of $d$ on the performances. We focus on the "others" dataset because it is the most heterogeneous. We adopt a slightly different measure of performance and use precision recall (see for example [4]) to measure performances in identifying network intrusions (the higher the curve, the better). We call the area under such curves the AUPR. The right panel of Figure 2 represents these results. First, the case $d = 1$, which corresponds to vanilla Mahalanobis distance as outlined in Section 3.3, gives poor performances. Second, the global performances rapidly increase with $d$ and then decrease and stabilize.

This suggests that $d$ can be used as a tuning parameter to control the "complexity" of $Q_{\mu,d}$. Indeed, $2d$ is the degree of the polynomial $Q_{\mu,d}$ and it is expected that more complex models will identify more diverse classes of examples as outliers. In our case, this means identifying regular traffic as outliers while it actually does not correspond to intrusions. In general, a good heuristic regarding the tuning of $d$ is to investigate performances on a well specified task in a preliminary experiment.

# 6 Future work

An important question is the asymptotic regime when $d \to \infty$. Current state of knowledge suggests that, up to a correct scaling, the limit of the Christoffel functions (when known to exist) involves an edge effect term, related to the support of the measure, and the density of $\mu$ with respect to Lebesgue measure, see for example [2] for the Euclidean ball. It also suggests connections with the notion of equilibrium measure in potential theory [17, 1, 7]. Generalization and interpretation of these results in our context will be investigated in future work.

Even though good approximations are obtained with low degree (at least in dimension 2 or 3), the approach involves the inversion of large ill conditioned Hankel matrices which reduces considerably the applicability for higher degrees and dimensions. A promising research line is to develop approximation procedures and advanced optimization and algebra tools so that the approach could scale computationally to higher dimensions and degrees.

Finally, we did not touch the question of statistical accuracy. In the context of empirical processes, this will be very relevant to understand further potential applications in machine learning and reduce the gap between the abstract orthogonal polynomial theory and practical machine learning applications.

## Acknowledgments

This work was partly supported by project ERC-ADG TAMING 666981, ERC-Advanced Grant of the *European Research Council* and grant number FA9550-15-1-0500 from the *Air Force Office of Scientific Research, Air Force Material Command*.

## Footnotes

[1]For the null polynomial, we use the convention that its degree is 0 and it is $\leq_{gl}$ smaller than all other monomials.

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
