[Supplementary Material · typicSOSFullConcat.pdf]

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

# A Additional examples

Figure 3: Empirical measure and level sets of $Q_{\mu,d}$ for various values of $d$ and various configurations and numbers of points $n$. The level set $\binom{p+d}{d}$, which corresponds to the average value of $Q_{\mu,d}$, is represented in red.

# B Proofs

We use the same notation as in the main text. We recall that Assumption 1.

**Assumption 1** $\mu$ *is a Borel probability measure on $\mathbb{R}^p$ with all its moments finite and $M_d(\mu)$ is positive definite for a given $d \in \mathbb{N}$.*

Lemma 2 and Theorem 2 are taken from the literature and we provide a proof for completeness.

## B.1 Proof of Lemma 1

First we show that the mapping $\mathbf{x} \mapsto Q_{\mu,d}(\mathbf{x})$ does not depend on the choice of a specific basis of $\mathbb{R}_d[X]$. Then we will deduce the affine invariance property.

**Lemma 3** *Let $\mathbf{w}_d(X)$ be an arbitrary basis of $\mathbb{R}_d[X]$ and let $R_{\mu,d} \in \mathbb{R}_d[X]$ be derived in the same way as $Q_{\mu,d}$ (see Definition 1), with $\mathbf{w}_d$ in place of $\mathbf{v}_d$. Then $Q_{\mu,d}(\mathbf{x}) = R_{\mu,d}(\mathbf{x})$ for all $\mathbf{x} \in \mathbb{R}^p$.*

**Proof :** Since $\mathbf{w}_d$ is a basis of $\mathbb{R}_d[X]$, there exists an invertible matrix $C \in \mathbb{R}^{s(d) \times s(d)}$ such that $\mathbf{w}_d(X) = C\mathbf{v}_d(X)$. We reproduce the computation of Definition 1 with this new basis. We write $N_d(\mu)$ the moment matrix computed with the polynomial basis $\mathbf{w}_d$. We have

$$
\begin{aligned}
N_d(\mu) &= \int_{\mathbb{R}^p} \mathbf{w}_d(\mathbf{x})\mathbf{w}_d(\mathbf{x})^T d\mu(\mathbf{x}) \\
&= \int_{\mathbb{R}^p} C\mathbf{v}_d(\mathbf{x})\mathbf{v}_d(\mathbf{x})^T C^T d\mu(\mathbf{x}) \\
&= C \int_{\mathbb{R}^p} \mathbf{v}_d(\mathbf{x})\mathbf{v}_d(\mathbf{x})^T d\mu(\mathbf{x}) C^T \\
&= CM_d(\mu)C^T,
\end{aligned}
$$

which leads to $N_d(\mu)^{-1} = C^{-T}M_d(\mu)^{-1}C^{-1}$. Using Definition 1, for all $\mathbf{x} \in \mathbb{R}^p$, we have

$$
\begin{aligned}
R_{\mu,d}(\mathbf{x}) &= \mathbf{w}_d(\mathbf{x})^T N_d(\mu)^{-1}\mathbf{w}_d(\mathbf{x}) \\
&= \mathbf{v}_d(\mathbf{x})^T C^T C^{-T} M_d(\mu)^{-1} C^{-1} C\mathbf{v}_d(\mathbf{x}) \\
&= \mathbf{v}_d(\mathbf{x})^T M_d(\mu)^{-1}\mathbf{v}_d(\mathbf{x}) \\
&= Q_{\mu,d}(\mathbf{x}),
\end{aligned}
$$

which concludes the proof. $\qquad\square$

**Lemma 1** *Let $\mu$ satisfy Assumption 1 and $A \in \mathbb{R}^{p \times p}, b \in \mathbb{R}^p$ define an invertible affine mapping on $\mathbb{R}^p$, $\mathcal{A}: \mathbf{x} \to A\mathbf{x}+b$. Then, the push foward measure, defined by $\tilde{\mu}(S) = \mu(\mathcal{A}^{-1}(S))$ for all Borel sets $S \subset \mathbb{R}^p$, satisfies Assumption 1 (with the same $d$ as $\mu$) and for all $\mathbf{x} \in \mathbb{R}^p$, $Q_{\mu,d}(\mathbf{x}) = Q_{\tilde{\mu},d}(A\mathbf{x}+b)$.*

**Proof :** Let us first compute $M_d(\tilde{\mu})$. For the push forward measure $\tilde{\mu}$, it holds that for any $\mu$ integrable function $f: \mathbb{R}^p \to \mathbb{R}$,

$$
\int_{\mathbb{R}^p} f(\mathbf{x})d\tilde{\mu}(\mathbf{x}) = \int_{\mathbb{R}^p} f(A\mathbf{x}+b)d\mu(\mathbf{x}).
$$

By considering polynomial $f$, we have that $\tilde{\mu}$ has all its moments finite and satisfies Assumption 1 with the same $d$ as $\mu$. Furthermore, we have

$$
M_d(\tilde{\mu}) = \int_{\mathbb{R}^p} \mathbf{v}_d(\mathbf{x})\mathbf{v}_d(\mathbf{x})^T d\tilde{\mu}(\mathbf{x}) = \int_{\mathbb{R}^p} \mathbf{v}_d(A\mathbf{x}+b)\mathbf{v}_d(A\mathbf{x}+b)^T d\mu(\mathbf{x}). \tag{7}
$$

We can deduce the following identity for all $\mathbf{x} \in \mathbb{R}^p$,

$$
Q_{\tilde{\mu},d}(A\mathbf{x}+b) = \mathbf{v}_d(A\mathbf{x}+b)^T M_d(\tilde{\mu})^{-1} \mathbf{v}_d(A\mathbf{x}+b). \tag{8}
$$

It remains to notice that mappings defined by $\mathbf{w}_d(\mathbf{x}) = \mathbf{v}_d(A\mathbf{x}+b)$ for all $\mathbf{x} \in \mathbb{R}^p$ form a basis of the polynomials of degree up to $d$ on $\mathbb{R}^p$ (by invertibility of the affine mapping). Combining (7) and (8), we see that $\mathbf{x} \mapsto \mathbf{v}_d(A\mathbf{x}+b)$ simply corresponds to the use of a different basis of $\mathbb{R}_d[X]$. The result follows by applying Lemma 3 and the proof is complete. $\qquad\square$

## B.2 Proof of Lemma 2

Recall that the orthogonal polynomials satisfy for all $\alpha \in \mathbb{N}_d^p$

$$\langle P_\alpha, X^\beta \rangle = 0 \text{ if } \alpha <_{gl} \beta, \ \langle P_\alpha, P_\alpha \rangle_\mu = 1, \ \langle P_\alpha, X^\beta \rangle_\mu = 0 \text{ if } \beta <_{gl} \alpha, \ \langle P_\alpha, X^\alpha \rangle_\mu > 0. \quad (3)$$

**Lemma 2** *Let $\mu$ satisfy Assumption 1, then $Q_{\mu,d} = \sum_{\alpha \in \mathbb{N}_d^p} P_\alpha^2$, where the family $\{P_\alpha\}_{\alpha \in \mathbb{N}_d^p}$ is defined by (3) and $\int_{\mathbb{R}^p} Q_{\mu,d}(\mathbf{x}) d\mu(\mathbf{x}) = s(d)$.*

**Proof :** Let $D_d(\mu)$ be the lower triangular matrix which rows are the coefficients of the polynomials $P_\alpha$ defined in (3) ordered by $\leq_{gl}$. From properties in (3), $D_d(\mu)$ is lower triangular with positive coefficients on its diagonal and therefore invertible. We have $D_d(\mu) M_d(\mu) D_d(\mu)^T = I$, the identity. It follows that $M_d(\mu) = D_d(\mu)^{-1} D_d(\mu)^{-T}$ and $M_d(\mu)^{-1} = D_d(\mu)^T D_d(\mu)$. Plugging this in definition 1 and using equation (1) leads to the desired identity. The average value result follows because we manipulate an orthonormal basis of $s(d)$ polymials, each of which has a square average value (with respect to $\mu$) equal to 1. $\qquad \square$

## B.3 Proof of Theorem 1

We recall the the optimization problem.

$$\min_{Q_\alpha, \theta_\alpha, \alpha \in \mathbb{N}_d^p} \frac{1}{2} \int_{\mathbb{R}^p} \sum_{\alpha \in \mathbb{N}_d^p} Q_\alpha(\mathbf{x})^2 d\mu(\mathbf{x}) \quad (4)$$

$$\text{s.t.} \ \ \mathbf{q}_{\alpha\alpha} \geq \exp(\theta_\alpha), \quad \alpha \in \mathbb{N}_d^p,$$
$$\mathbf{q}_{\alpha\beta} = 0, \ \alpha <_{gl} \beta, \quad \alpha, \beta \in \mathbb{N}_d^p,$$
$$\sum_{\alpha \in \mathbb{N}_d^p} \theta_\alpha = 0.$$

where $Q_\alpha(\mathbf{x}) = \sum_\beta \mathbf{q}_{\alpha\beta} \mathbf{x}^\beta, \alpha \in \mathbb{N}_d^p$. The statement of Theorem 1 goes as follows.

**Theorem 1** *: Problem (4) is a convex optimization problem with a unique optimal solution $(Q_\alpha^*, \theta_\alpha^*)$, which satisfies $Q_\alpha^* = \sqrt{\lambda} P_\alpha, \ \alpha \in \mathbb{N}_d^p$, for some $\lambda > 0$. In particular, the distinguished SOS polynomial*

$$Q_{\mu,d} = \sum_{\alpha \in \mathbb{N}_d^p} P_\alpha^2 = \frac{1}{\lambda} \sum_{\alpha \in \mathbb{N}_d^p} (Q_\alpha^*)^2,$$

*is (part of) the unique optimal solution of (4).*

**Proof :**

**General remarks.** Observe that (4) is a convex optimization problem as we have $\int_{\mathbb{R}^p} \sum_{\alpha \in \mathbb{N}_d^p} Q_\alpha(\mathbf{x})^2 d\mu(\mathbf{x}) = \sum_{\alpha \in \mathbb{N}_d^p} \mathbf{q}_\alpha^T M_d(\mu) \mathbf{q}_\alpha$, which is strictly convex in $\{\mathbf{q}_\alpha\}_{\alpha \in \mathbb{N}_d^p}$. The proof is based on KKT optimality conditions for Problem (4). We first prove that any optimal solution should be of the form $Q_\alpha^* = \sqrt{\lambda} P_\alpha, \ \alpha \in \mathbb{N}_d^p$, for some $\lambda > 0$. Then we show that there exists a solution of the KKT system which has this form and finally that this solution is unique. The conclusion of Theorem 1 will then follow from Lemma 1. We begin with some notations that we will use throughout the proof.

**Notation.** Let $\{\mathbf{e}_\alpha\}_{\alpha \in \mathbb{N}_d^p}$ denote the canonical basis of $\mathbb{R}^{s(d)}$ indexed by $\alpha \in \mathbb{N}_d^p$ according to $\leq_{gl}$ order. The orthonormal polynomials $\{P_\alpha\}_{\alpha \in \mathbb{N}_d^p}$ (with respect to $\mu$) are uniquely defined. For each $\alpha \in \mathbb{N}_d^p$, we write $\mathbf{p}_\alpha = (\mathbf{p}_{\alpha\beta})_{\beta \in \mathbb{N}_d^p} \in \mathbb{R}^{s(d)}$ the coefficients of the polynomial $P_\alpha$. By construction of $P_\alpha$, for every $\alpha, \beta \in \mathbb{N}_d^p, \alpha <_{gl} \beta, \mathbf{p}_{\alpha\beta} = 0$ and $\mathbf{p}_{\alpha\alpha} > 0$.

**Optimality conditions** Problem (4) is strictly feasible, we can choose any $\theta$ such that $\sum_\alpha \theta_\alpha = 0$ and for every $\alpha \in \mathbb{N}_d^p$, set $Q_\alpha := \kappa P_\alpha$ for some sufficiently large $\kappa > 0$. Therefore the KKT optimality conditions are necessary and sufficient for global optimality. We introduce Lagrange multipliers for problem (4): $\lambda_\alpha \geq 0$ for each inequality constraint, $\lambda_{\alpha\beta} \in \mathbb{R}$ for each linear equality constraint on polynomials with $\alpha <_{gl} \beta$ and $\lambda \in \mathbb{R}$ for the last linear equality constraint on $\{\theta_\alpha\}_{\alpha \in \mathbb{N}_d^p}$. The KKT optimality conditions for problem (4) can be written as follows

$$\lambda_\alpha \geq 0, \quad \mathbf{e}_\alpha^T \mathbf{q}_\alpha^* \geq \exp(\theta_\alpha^*), \quad \alpha \in \mathbb{N}_d^p, \tag{9}$$

$$\mathbf{e}_\beta^T \mathbf{q}_\alpha^* = 0, \quad \alpha, \beta \in \mathbb{N}_d^p, \quad \alpha <_{gl} \beta, \tag{10}$$

$$\sum_{\alpha \in \mathbb{N}_d^p} \theta_\alpha^* = 0, \tag{11}$$

$$M_d(\mu)\mathbf{q}_\alpha^* = \lambda_\alpha \mathbf{e}_\alpha + \sum_{\alpha <_{gl} \beta} \lambda_{\alpha\beta}\, \mathbf{e}_\beta, \quad \alpha \in \mathbb{N}_d^p, \tag{12}$$

$$\lambda_\alpha \exp(\theta_\alpha^*) = \lambda_\alpha \mathbf{e}_\alpha^T \mathbf{q}_\alpha^* = \lambda, \quad \alpha \in \mathbb{N}_d^p, \tag{13}$$

for optimal variables $\theta_\alpha^*$, polynomials $Q_\alpha^*$ with coefficients $\mathbf{q}_\alpha^* \in \mathbb{R}^{s(d)}$, for each $\alpha \in \mathbb{N}_d^p$. We next show that the part $(Q_\alpha^*)_{\alpha \in \mathbb{N}_d^p}$ of an optimal solution is necessarily a family of orthogonal polynomials.

**Any optimal solution has the form $Q_\alpha^* = \sqrt{\lambda}P_\alpha$, $\alpha \in \mathbb{N}_d^p$, for some $\lambda > 0$.** Since KKT conditions are necessary and sufficient for optimality, we only focus on them. For each $\alpha \neq 0$ and $\beta <_{gl} \alpha$, multiplying (12) by $\mathbf{e}_\beta$, we obtain

$$\langle X^\beta, Q_\alpha^* \rangle_\mu = \int \mathbf{x}^\beta Q_\alpha^*(\mathbf{x})\, d\mu(\mathbf{x}) = \mathbf{e}_\beta^T M_d(\mu)\, \mathbf{q}_\alpha^* = \lambda_\alpha\, \mathbf{e}_\beta^T \mathbf{e}_\alpha + \sum_{\alpha <_{gl} \gamma} \lambda_{\alpha\gamma}\, \mathbf{e}_\beta^T \mathbf{e}_\gamma = 0. \tag{14}$$

Similarly, multiplying (12) by $\mathbf{q}_\alpha^*$ yields for all $\alpha \in \mathbb{N}_d^p$,

$$\langle Q_\alpha^*, Q_\alpha^* \rangle_\mu = \int Q_\alpha^*(\mathbf{x})^2\, d\mu(\mathbf{x}) = (\mathbf{q}_\alpha^*)^T M_d(\mu)\, \mathbf{q}_\alpha^* = \lambda_\alpha\, (\mathbf{q}_\alpha^*)^T \mathbf{e}_\alpha = \lambda, \tag{15}$$

where we have used (13) for the last identity. In particular, with $\alpha = 0$, $Q_0^*(\mathbf{x}) = \mathbf{q}_{00}^* (\geq \exp(\theta_0^*))$ for all $\mathbf{x}$ and so

$$\lambda = \int Q_0^*(\mathbf{x})^2\, d\mu(\mathbf{x}) = (\mathbf{q}_{00}^*)^2 \int d\mu \geq \exp(2\theta_0^*),$$

which shows that $\lambda > 0$. Next, combining (14), (15) and the condition (10), we immediately deduce

$$\langle Q_\beta^*, Q_\alpha^* \rangle_\mu = \int Q_\beta^*(\mathbf{x})\, Q_\alpha^*(\mathbf{x})\, d\mu(\mathbf{x}) = \begin{cases} \lambda & \text{if } \alpha = \beta \\ 0 & \text{otherwise.} \end{cases} \tag{16}$$

Finally, for every $\alpha \in \mathbb{N}_d^p$, multiplying (12) by $\mathbf{e}_\alpha$ yields

$$\langle X^\alpha, Q_\alpha^* \rangle_\mu = \int \mathbf{x}^\alpha Q_\alpha^*(\mathbf{x})\, d\mu(\mathbf{x}) = \mathbf{e}_\alpha^T M_d(\mu)\, \mathbf{q}_\alpha^* = \lambda_\alpha > 0, \quad \alpha \in \mathbb{N}_d^p. \tag{17}$$

The last inequality follows from (13). Indeed, suppose $\lambda_\alpha = 0$ for some $\alpha \in \mathbb{N}_d^p$, this would yield $\lambda = 0$. Since we have shown that $\lambda > 0$, it must also hold that $\lambda_\alpha > 0$ for all $\alpha$. Combining relations (10), (14), (16) and (17), we have shown that the $\{Q_\alpha^*\}_{\alpha \in \mathbb{N}_d^p}$ form a family of orthogonal polynomials with respect to $\mu$. In addition, by the uniqueness of the orthonormal basis $\{P_\alpha\}_{\alpha \in \mathbb{N}_d^p}$, it follows from (16) that $Q_\alpha^* = \sqrt{\lambda}\, P_\alpha$ for every $\alpha \in \mathbb{N}_d^p$.

**There exists a solution of this form.** Recall that, for each $\alpha \in \mathbb{N}_d^p$, $\mathbf{p}_\alpha = (\mathbf{p}_{\alpha\beta})_{\beta \in \mathbb{N}_d^p} \in \mathbb{R}^{s(d)}$ is the vector of coefficients of the polynomial $P_\alpha$ which satisfies by construction $\mathbf{p}_{\alpha\alpha} > 0$ and $\mathbf{p}_{\alpha\beta} = 0$

for all $\beta \in \mathbb{N}_d^p$, $\alpha <_{gl} \beta$. We use the following assignment for the primal and dual variables.

$$\lambda = \left( \prod_{\alpha \in \mathbb{N}_d^p} \mathbf{p}_{\alpha\alpha} \right)^{\frac{-2}{s(d)}} > 0 \tag{18}$$

$$\lambda_\alpha = \sqrt{\lambda} \mathbf{e}_\alpha^T M_d(\mu) \mathbf{p}_\alpha = \frac{\sqrt{\lambda}}{\mathbf{p}_{\alpha\alpha}} > 0, \quad \alpha \in \mathbb{N}_d^p$$

$$\lambda_{\alpha\beta} = \sqrt{\lambda} \mathbf{e}_\beta^T M_d(\mu) \mathbf{p}_\alpha, \quad \alpha, \beta \in \mathbb{N}_d^p, \alpha <_{gl} \beta$$

$$\mathbf{q}_\alpha^* = \sqrt{\lambda} \mathbf{p}_\alpha, \quad \theta_\alpha^* = \log(\sqrt{\lambda} \mathbf{p}_{\alpha\alpha}), \quad \alpha \in \mathbb{N}_d^p.$$

Using orthonormality of the polynomials $\{P_\alpha\}_{\alpha \in \mathbb{N}_d^p}$, it can be check that the assignment (18) satisfies KKT optimality conditions (9), (10), (11), (12) and (13). We have therefore constructed an optimal solution of (4) with the desired form.

**The optimal solution is unique.** From what precedes any optimal solution of (4) is necessarily such that $Q_\alpha^* = \sqrt{\lambda} P_\alpha$, for every $\alpha \in \mathbb{N}^n$, for some $\lambda > 0$. In addition the optimal value of (4) is $s(d)\lambda$. Suppose that there exists two different optimal solutions $(Q_\alpha, \theta_\alpha)_{\alpha \in \mathbb{N}_d^p}$ and $(Q'_\alpha, \theta'_\alpha)_{\alpha \in \mathbb{N}_d^p}$ with associated dual variables $(\lambda, \lambda_\alpha, \lambda_{\alpha\beta})_{\alpha,\beta \in \mathbb{N}_d^p}$ and $(\lambda', \lambda'_\alpha, \lambda'_{\alpha\beta})_{\alpha,\beta \in \mathbb{N}_d^p}$. Then necessarily $\lambda = \lambda'$, $Q_\alpha = Q'_\alpha = \sqrt{\lambda} P_\alpha$ and $\lambda_\alpha, \lambda'_\alpha > 0$ for all $\alpha \in \mathbb{N}_d^p$. But then from (13), $\sqrt{\lambda} \mathbf{p}_{\alpha\alpha} = \exp(\theta_\alpha) = \exp(\theta'_\alpha)$ and so $\theta'_\alpha = \theta_\alpha$ for every $\alpha \in \mathbb{N}_d^p$. Therefore the solution is unique and this concludes the proof of Theorem 1. $\qquad \square$

## B.4 Proof of Theorem 2

**Theorem 2** *Let Assumption 1 hold and let $\mathbf{z} \in \mathbb{R}^p$ be fixed, arbitrary. Then*

$$Q_{\mu,d}(\mathbf{z})^{-1} = \min_{P \in \mathbb{R}_d[X]} \left\{ \int P(\mathbf{x})^2 \, d\mu : P(\mathbf{z}) = 1 \right\}. \tag{7}$$

**Proof :**

Fix an arbitrary $P \in \mathbb{R}_d[X]$ and $\mathbf{z} \in \mathbb{R}^p$. Assume that $P(\mathbf{z}) = 1$. Letting for all $\alpha \in \mathbb{N}_d^p$, $a_\alpha = \langle P, P_\alpha \rangle_\mu$, by orthonormality, we have

$$P = \sum_{\alpha \in \mathbb{N}_d^p} a_\alpha P_\alpha, \tag{19}$$

$$\langle P, P \rangle_\mu = \sum_{\alpha \in \mathbb{N}_d^p} a_\alpha^2.$$

The assumption that $P(\mathbf{z}) = 1$ can be used in conjonction with Cauchy-Schwartz inequality to obtain

$$1 = P(\mathbf{z}) \tag{20}$$

$$= \sum_{\alpha \in \mathbb{N}_d^p} a_\alpha P_\alpha(\mathbf{z})$$

$$\leq \left( \sum_{\alpha \in \mathbb{N}_d^p} a_\alpha^2 \right) \left( \sum_{\alpha \in \mathbb{N}_d^p} P_\alpha(\mathbf{z})^2 \right)$$

$$= \langle P, P \rangle_\mu Q_{\mu,d}(\mathbf{z}),$$

where the last equality comes from the definition of $\langle \cdot, \cdot \rangle_\mu$ and Lemma 2. There is equality in equation (20) if and only if $a_\alpha = P_\alpha(\mathbf{z})/Q_{\mu,d}(\mathbf{z})$ which always leads to $P(\mathbf{z}) = 1$. This shows that the infimum is attained and concludes the proof. $\qquad \square$

## C   Details about the preparation of the datasets

We reproduce the exact same manipulations as in the references. We downloaded the `kddcup.data` from the following repository `https://archive.ics.uci.edu/ml/machine-learning-databases/kddcup99-mld/`. This file contains $4898431$ instances of network connections described by 42 features including the type of connection (attack or normal). We filter the records by keeping only those for which the variable *logged in* is positive. We kept the labels (type of connection: intrusion or not) together with the four most important features: *service, duration, src_bytes, dst_bytes*. We applied to the three last variables (numerical) the function $\log(\cdot + 0.1)/10$. We build four datasets with the four most frequent instances of *service* and group all the remaining records in the dataset *others* to get our five datasets, each of them described by three variables as the *service* is used to differentiate the datasets.

## D   Illustration of affine invariance

The following Figure illustrate the affine invariance property described in Lemma 1.

Figure 4: Empirical measure and level sets of $Q_{\mu,d}$ for three configurations of the same cloud of points ($d = 4$). The cloud in the middle is rotation of the original one and the cloud on the right is the same after centering and scaling. We observe that the level sets follow the same transformations.

## E   Illustration of the extremal property of the Christoffel function

In order to illustrate the discussion of Section 4.4 regarding Theorem 2, we represent the solution of the optimization problem (7) in a concrete setting. We consider the cloud of points represented in Figure 5 (501 points, including the red one, an outlier) and set $\mathbf{z}$ to be the red point. We choose $d = 5$, and $\mu$ the empirical average of the 501 points. The value of $Q_{\mu,d}(\mathbf{z})$ is around 420. Let $P$ be the solution of problem (7), the proof of Theorem 2 shows how to compute this solution. The discussion of Section 4.4 ensures that the set $\left\{ \mathbf{x} \in \mathbb{R}^2 : P(\mathbf{x})^2 \leq 1 \right\}$ contains at least $1 - 1/420$ of the mass of the empirical measure. Since there are $501$ points, this means that this sublevel set contains at least 500 of the 501 original points. Hence we have a certificate that there exists a polynomial of degree 10 for which 500 out of the 501 points have value less or equal to 1 while the value at $\mathbf{z}$ is 1. Figure 5 shows the level sets of $P^2$, $\mathbf{z}$ indeed belongs to the level set 1, and actually, all the points are contained in the red sublevel set (more than predicted by theory).

Figure 5: 501 points and level sets of $P^2$ (red is $P^2 = 1$), where $P$ is a solution of problem (7) in Theorem 2 ($d = 5$). $\mathbf{z}$ is the red point on the figure and the value of $Q_{\mu,d}(\mathbf{z})$ is approximately 420.