[Reviews · NeurIPS 2016]

Reviewer 1

Summary

In this submission, the author(s) provide(s) an interesting use of the Christoffel function for important topics such as outlier detection or shape reconstruction from discrete obervations. This tool is new and seems promising. The paper gives some fundamental properties of this function such as SoS representability and affine invariance. The most powerful feature remains that this function is somehow a "natural" polynomial approximation of the indicator function of the support of the (unknown) population measure (see Theorem 2). To my point of view, this property should be emphasized in the revision. Ideally, a convergence result of the christoffel function towards the support indicator function should be mentioned or proved.

Qualitative Assessment

Is there any hope to prove that the Christoffel function converges (point-wise or in a stronger sense) towards the indicator function of the support measure as n and d tend to infinity?

Confidence in this Review

3-Expert (read the paper in detail, know the area, quite certain of my opinion)


Reviewer 2

Summary

This paper makes an empirical observation that a certain polynomial which has the form of Sum-of-Squares can capture the shape of point clouds. The polynomial is based on the moments of the point distribution, and the paper gives some justifications. The method is applied to network intrusion data sets and the results seem reasonable.

Qualitative Assessment

The main idea of the paper is to find a polynomial that corresponds to the "whitening" transformation of the high-order moment Hankel matrix. The simplest case is when the Hankel matrix is just the covariance matrix and this then corresponds to "whiten" the data (putting the data into isotropic position). However as the paper also mentiones that it is not very clear how this intuition can be generalized to higher order. Figure 1 illustrates really good performance but both examples are low dimension and it feels like the distribution can be well approximated by low-degree polynomial. It would be good if the paper can offer more insights into what this transformation corresponds to in higher dimensional cases. The paper characterizes the polynomial using a certain variational formulation, it is reasonable, but at the same time the objective function is not particularly natural and feels like it is chosen in order to make the variational formulation work. Assumption 1 assumes M_d(\mu) is positive definite, which is reasonable (the matrix is always PSD and should be positive definite as long as the distribution is not precisely described using some polynomial equations). However, even if the matrix is positive definite it may still have a very small singular value, which will make the inverse matrix unstable and might potentially hurt the polynomial. The paper should comment on whether that happens for some larger d in practice or not. The experiments is interesting and the results look reasonable. The reviewer is not very familiar with this data set and it would be good to have comparisons with some standard methods. Overall this is an interesting paper.

Confidence in this Review

2-Confident (read it all; understood it all reasonably well)


Reviewer 3

Summary

This is a well-written paper that describes the construction of a polynomial whose sublevel sets turn out to be strikingly aligned with the shape of a given point cloud of data. The construction involves computing the inverse of a specific moment matrix and using it to define a sum-of-squares polynomial. From a practical point of view, the method is highly scalable (linear) in the number of data points, though it scales exponentially in terms of dimensionality. It could definitely find applications in ML.

Qualitative Assessment

Very nice paper. The proposed procedure could be of value in anomaly detection and shape modeling problems, particularly since it scales well with respect to number of data points. - Once the polynomial in Eqn 2 is constructed, is the sublevel set value of (p+d)-choose-d always guaranteed to contain the points? Why (p+d)-choose-d and can that value somehow be normalized (say 1)? - Is there a heuristic you would suggest to pick the degree with which to model the shape? - In what precise sense is the polynomial in Eqn 2 right fitting? Does it optimize some explicit notion of volume (analogous to determinant for ellipsoids)? - It would be interesting to study the degree=2 case and its relationship to minimum volume ellipsoids. - Typo below Eqn 3: "...lower triangular matrix which rows"--> "...lower triangular matrix whose rows" - Can the form of the family of polynomials in Eqn 3 be made explicit? - I found the intuition around Eqn 4 bit hard to digest. Please expand the section and try to make it as clear as possible since its in some sense the theory that explains the phenomenon. I found Theorem 2 to be easier to appreciate.

Confidence in this Review

3-Expert (read the paper in detail, know the area, quite certain of my opinion)


Reviewer 4

Summary

Given an empirically observed that the sublevel sets of a certain distinguished polynomial capture the shape of the cloud very accurately, a mathematical rationale for this observation is studied to represent a cloud of data points with the inverse moment matrix SOS polynomial

Qualitative Assessment

Given an empirically observed that the sublevel sets of a certain distinguished polynomial capture the shape of the cloud very accurately, a mathematical rationale for this observation is studied to represent a cloud of data points with the inverse moment matrix SOS polynomial. The comments are as follows 1.The SOS polynomial representation comes from some empirical observation; however, in this paper, it seems that the empirical observation is not described sufficiently. 2.It is necessary to demonstrate the superiority and effectiveness of the proposed method with comparison with previous methods. 3.This paper is not organized enough. For a reader, we have to find the meaning of \theta_\alpha in (4) in the proof?

Confidence in this Review

1-Less confident (might not have understood significant parts)


Reviewer 5

Summary

The authors use the inverse of the moment matrix (up to a certain degree) and connect its inverse to a family of polynomial functions, which are used for outlier detection.

Qualitative Assessment

The theory developed (and other connections, e.g. variational objective) is interesting. The approach seems to be novel; although my biggest criticism to this work is the unclarity of some ideas presented. If we consider that the reader of this paper will be an general "learning theory" community, I think some of the keywords need to be clearly defined (e.g. "Orthogonal polynomials"); they might be popular in math community, but I don't think they are clear to average learning community. In places giving more intuition would definitely help (for example Sec4.4 which has details of outlier detection) Also I wish you had examples explaining giving more intuition the orthogonal polynomials, the Christoffel, etc. Note that some of the above criticisms might be because I don't have the same background as yours. Side question: Any thoughts on using your transformations for the clustering problem? Minor: - Line 71: “ use the canonical order … ” → using? - “a sequence of efficient Woodbury updates” → add a reference or be more precise what you mean by this. - Experiments: it would have been great if the authors reported the baseline scores for completeness.

Confidence in this Review

1-Less confident (might not have understood significant parts)


Reviewer 6

Summary

This paper uses sum-of-squared (SOS) polynomials (i.e. the set of polynomials of degree 2d which can be written as the square of polynomials of degree d) to detect outliers in point sets. In contrast to kernel-based methods such as Gaussian processes, the evaluation of an SOS polynomials depends on the number of monomials (and thus on the ambient dimension) rather than on the number of observations. Thus the approach seems obviously advantageous for evaluating typicality in very large point sets of relatively low dimensionality. The paper provides some nice theoretical analysis that helps to explain empirically-observed properties of SOS polynomials that make them useful for typicality. Finally, the authors apply the technique to a dataset of network

Qualitative Assessment

This paper is clear and seems thorough and technically sound, at least up to the depth at which I could follow it. If it is indeed the case that the presented empirical observation of the utility of SOS polynomials for describing typicality in point sets has not been previously noted in the literature, then it would be a clearly useful contribution. The experimental results show that the SOS polynomial technique is competitive with other techniques hand-designed to detect typicality while also being simple and well-justified from a theoretical standpoint. My only concern is regarding the fitness of the presented work for presentation in NIPS. It seems that this approach would be extremely useful for geometric settings in which the dimensionality is low, e.g. 2 or 3, but the number of points can be in the millions or more, perhaps for 3D point-cloud de-noising applications. As for machine learning, the paper presents one example that seems reasonable, but it would be useful if some additional examples of where this approach my prove advantageous were given, or even just some additional motivation in the text describing potential applications. Minor points on clarity: I assume x bar in section 4.4 (introduced in line 239) is referring as usual to the mean of a set of points in R^p but that wasn't made entirely clear and I don't think it was used previously. On line 257 in section 5 it says that the network intrusion dataset consists of labelled vectors in R^3, but in lines 441-442 of Appendix C it says "the four most important features" are used. Is this a typo or is there some sort of projection from the four features to a 3D space being used? A very minor point, but it seems that the section describing affine invariance is really describing affine co-variance, i.e. if you transform the point set using an affine transformation, the resulting SOS is different (thus not invariant) but it is equivalent to transforming the underlying coordinate system of the polynomial (thus co-variant).

Confidence in this Review

1-Less confident (might not have understood significant parts)